# Response Surface Optimisation of Polydimethylsiloxane (PDMS) on Borosilicate Glass and Stainless Steel (SS316) to Increase Hydrophobicity

**DOI:** 10.3390/molecules27113388

**Published:** 2022-05-25

**Authors:** Nadiah Ramlan, Saiful Irwan Zubairi, Mohamad Yusof Maskat

**Affiliations:** 1Academy of Contemporary Islamic Studies, Universiti Teknologi Mara (UiTM), Shah Alam 40450, Malaysia; nadiahramlan@uitm.edu.my; 2Department of Food Sciences, Faculty of Science & Technology, Universiti Kebangsaan Malaysia, Bangi 43600, Malaysia; yusofm@ukm.edu.my; 3Tasik Chini Research Centre (PPTC), Faculty of Science & Technology, Universiti Kebangsaan Malaysia, Bangi 43600, Malaysia

**Keywords:** optimisation, surface modification, contact angle, PDMS, stainless steel, borosilicate glass

## Abstract

Particle deposition on the surface of a drying chamber is the main drawback in the spray drying process, reducing product recovery and affecting the quality of the product. In view of this, the potential application of chemical surface modification to produce a hydrophobic surface that reduces the powder adhesion (biofouling) on the wall of the drying chamber is investigated in this study. A hydrophobic polydimethylsiloxane (PDMS) solution was used in the vertical dipping method at room temperature to determine the optimum coating parameters on borosilicate glass and stainless steel substrates, which were used to mimic the wall surface of the drying chamber, to achieve highly hydrophobic surfaces. A single-factor experiment was used to define the range of the PDMS concentration and treatment duration using the Response Surface Methodology (RSM). The Central Composite Rotatable Design (CCRD) was used to study the effects of the concentration of the PDMS solution (X_1_, %) and the treatment duration (X_2_, h) on the contact angle of the substrates (°), which reflected the hydrophobicity of the surface. A three-dimensional response surface was constructed to examine the influence of the PDMS concentration and treatment duration on contact angle readings, which serve as an indicator of the surface’s hydrophobic characteristics. Based on the optimisation study, the PDMS coating for the borosilicate glass achieved an optimum contact angle of 99.33° through the combination of a PDMS concentration of X_1_ = 1% (*w*/*v*) and treatment time of X_2_ = 4.94 h, while the PDMS coating for the stainless steel substrate achieved an optimum contact angle of 98.31° with a PDMS concentration of X_1_ = 1% (*w*/*v*) and treatment time of X_2_ = 1 h. Additionally, the infrared spectra identified several new peaks that appeared on the PDMS-treated surfaces, which represented the presence of Si-O-Si, Si-CH_3_, CH_2_, and CH_3_ functional groups for the substrates coated with PDMS. Furthermore, the surface morphology analysis using the Field Emission Scanning Electron Microscopy (FESEM) showed the presence of significant roughness and a uniform nanostructure on the surface of the PDMS-treated substrates, which indicates the reduction in wettability and the potential effect of unwanted biofouling on the spray drying chamber. The application of PDMS and PTFE on the optimally coated substrates successfully reduced the amount of full cream milk particles that adhered to the surface. The low surface energy of the treated surface (19–27 mJ/m^2^) and the slightly higher surface tension of the full cream milk (54–59 mJ/m^2^) resulted in a high contact angle (102–103°) and reduced the adhesion work on the treated substrates (41–46 mJ/m^2^) as compared to the native substrates.

## 1. Introduction

A spray dryer is a frequently used equipment in the food industry to convert a liquid sample into powder form. Given its significant importance, the reduced operating efficiency and changes in the product yield and quality due to powder deposition on the spray dryer are major concerns that need to be addressed. The high temperature during the spray drying process also increases the tendency of the naturally amorphous state of the food to become sticky, which is described as the glass transition phenomenon. Thus, deposit removal is considered a significant economic and environmental burden to the industry [1]. Although several studies have been conducted to address this issue, including the use of carrier agents and controlling processing parameters [2,3], these methods lead to new problems that alter the quality and characteristics of the dried product.

Furthermore, the properties of the wall of the spray dryer on which the powder adhesion takes place play a crucial role in the product deposition mechanism. Generally, fluid adhesion on the wall of the dryer depends on the surface properties of the wall [4]. A previous study by Woo et al. (2009) [5] reported that the wall material significantly affects the rate of particle deposition. Moreover, the wettability properties, which demonstrate the type of interaction between the solid surface and different types of liquids, have been investigated to produce anti-stick surfaces that facilitate the cleaning of the surface, thus lowering the cost of cleaning tools and the price of the product adhesion. The nature of the wettability is expressed directly through the liquid contact angle readings on the surface of the material [6]. A contact angle above 90° indicates that the surface material is hydrophobic with low wettability properties. Theoretically, the contact angle for a hydrophobic surface should not be more than 120° [7].

Meanwhile, silicon is a polymer that contains a siloxane bond (Si-O-Si). The terms silicon and siloxane are often interchangeable to represent the same material. While the number of siloxane bonds could vary depending on the number of organic groups and the number of oxygen atoms, siloxane can have an empirical formula ranging from (R_3_Si)_2_O to (RSiO_1.5_)_x_ [8] with silicon polymers having the general formula of (RR’SiO)_x_. When the R and R’ are from the methyl group, the silicon is specifically referred to as polydimethylsiloxane (PDMS). Recently, siloxane, which is the functional group of silicon, has been identified as one of the most promising materials that provides great hydrophobic coating due to the presence of the siloxane spinal network [9].

In this research, a simple surface modification treatment method is proposed to form hydrophobic surfaces through the use of a hydrophobic organic PDMS solution without any additional material at room temperature. The present study aims to elucidate the optimum conditions for the hydrophobic surface coating process using PDMS on borosilicate glass and stainless steel substrates. This simple surface modification by using the vertical dipping method was carried out to maximise the PDMS features and decrease the powder adhesion on the spray dryer wall without the use of fluorine materials and the intense drying temperatures typically used to provide hydrophobic surfaces [10]. The optimised parameters (duration of the coating process and the concentration of the PDMS solution) for the substrate surface coating process were determined using the Response Surface Methodology (RSM) method based on the resulting contact angle values. Subsequently, the surface morphology of the optimally PDMS-treated substrates was analysed using the Field Emission Scanning Electron Microscope (FESEM), while the presence of functional groups was determined using the Fourier Transform Infrared-Attenuated Total Reflectance (FTIR-ATR). Finally, the PDMS-treated surfaces were analysed by contact angle, surface energy, and work of adhesion by using raw whole milk product. 

## 2. Materials and Methods

### 2.1. Chemicals and Materials

The chemicals used in this study include Dow Corning MDX4-4159 50% Medical Grade Dispersion containing 50% active silicon dimethylsiloxane copolymer with functional amino acids in a mixture of aliphatic and isopropanol solvent (Dow Corning, Midland, MI, USA) and hexane 95% (*v*/*v*) (Merck, Darmstadt, Germany). Meanwhile, borosilicate glass microscope slides (dimension: 25 mm (h) × 25 mm (w) × 1 mm (t)) (Quasi-S Technology Sdn. Bhd., Bangi, Malaysia) and stainless steel slides (SS316) (dimension: 25 mm (h) × 25 mm (w) × 1.2 mm (t)) (Ikhlas Resmi (M) Sdn. Bhd., Shah Alam, Malaysia) were used as the substrates to mimic the pilot- (Buchi Mini Spray Dryer B-290, Büchi, Flawil, Switzerland) and industrial-scale spray drying chamber wall. The two substrates were chosen as they were frequently used either at the laboratory or industrial grade to study the powder deposition on both types of material [5,11]. 

### 2.2. Experimental Design of Surface Coating via the Central Composite Rotatable Design (CCRD)

The Dow Corning MDX4-4159 50% Medical Grade Dispersion containing 50% active silicon dimethylsiloxane with functional amino acids was diluted to an appropriate working concentration of 10% (*v*/*v*) using a hexane solution. The optimisation of the surface coating conditions was carried out by employing the Response Surface Methodology (RSM) using the Central Composite Rotatable Design (CCRD) with two main independent variables (X_1_: PDMS concentration (%, *w*/*v*) and X_2_: treatment time (h)), which was generated autonomously using a Design-Expert software version 6.0 (Table 1). As presented in Figure 1, the vertical dipping method at room temperature was employed based on the method by Wang et al. (2012) [10] with slight modifications. Following the surface modification treatment, the substrate was transferred into an oven dryer at 180 °C to evaporate the carrier solution and cure the coating materials on the substrate surface prior to the contact angle analysis. The experimental data were fitted with statistical models to produce the response surface. The models were deemed suitable when the one-way analysis of variance (ANOVA) was significant, the lack-of-fit test was insignificant, and the coefficient determination (R^2^) was more than 0.75 [12]. The chosen models were subsequently optimised based on the optimisation criteria of the minimum PDMS concentration and treatment time, while the contact angle was set to a maximum value to achieve a superhydrophobic effect [13].

### 2.3. Dairy Product Preparation 

Raw whole milk was purchased from Dutch Lady Milk Industries Berhad (Petaling Jaya, Malaysia). Dairy product compositions were determined based on previous reports [13,14] and are reported in Table 2. 

### 2.4. Water Contact Angle Measurement

The water contact angle measurement was performed on the surface of the borosilicate glass and stainless steel substrates using a Drop Shape Analyser DSA25E (Krüss, Hamburg, Germany) based on Sari (2006) [15] with slight modifications, as shown in Figure 2. The average contact angle values were determined at five different points on the surface of the substrate (*n* = 5) [16,17,18]. All measurements were carried out at room temperature (28 ± 2 °C). 

### 2.5. Surface Tension of Milk Products 

The surface tension of the liquid product used, full cream milk (*γ_LV_*), was calculated by first calculating the surface energy of the two substrates (*γ_SV_*) used, stainless steel and borosilicate glass. The Newmann curve was derived from the contact angle of the test liquid, whose surface tension was known, as measured on the substrate’s surface. To verify reproducibility, this contact angle value was measured three times for each kind of fluid. Deionized water (72.8 mJ/m^2^), glycerol (64 mJ/m^2^), and ethylene glycol (48 mJ/m^2^) were used as test liquids [1,17]. The surface temperatures of the treated and untreated stainless steel, as well as borosilicate glass, were maintained at room temperature. After determining the contact angle of the full cream milk product experimentally, the surface tension of the full cream milk liquid product was estimated using Equation (1) [19]:(1)$$\cos\theta =-1+2\sqrt{\frac{\gamma_{SV}}{\gamma_{LV}}}e^{-\beta {\left(\gamma_{LV}-\gamma_{SV}\right)}^{2}}$$

*θ* = contact angle (°);

*γ_SV_* = surface energy of the solid (mJ/m^2^);

*γ_LV_* = liquid surface tension (mJ/m^2^);

*β* = constants (mJ/m^2^)^−2^.

From a set of values of *γ_LV_* and *θ* measured using different types of reference fluids on a given solid surface, the values of the constants *β* and *γ_SV_* were determined using multi-variable optimization via the least-squares technique [19]. Starting with any value for the constants *β* and *γsv*, the iteration method can be used to identify the values of the pairs *β* and *γ_SV_* that best match Equation (1) to the experimental values of *γ_LV_* and *θ*.

### 2.6. The Work of Adhesion for Milk Products 

Adhesion work is defined as the amount of energy necessary to pull apart two in-contact phases inside a third phase for each unit of area [1,17]. It is estimated as an indication of the adhesion strength of various product powders on the spray dryer’s wall surface. The Young and Dupré equations can be used to compute adhesion work *W_a_*:(2)$$W_{a}=\gamma_{LV}\left(1+\mathrm{kos}\theta \right)$$

*θ* = contact angle (°);

*γ_LV_* = liquid surface tension (mJ/m^2^).

The adhesion strength of the liquid product on the surface of the substrate was calculated through the value of the contact angle that was obtained experimentally, as well as the surface tension value of the surface of the full cream milk on the substrate from Equation (1).

### 2.7. Morphological Surface Analysis: Field Emission Scanning Electron Microscopy (FESEM)

A Field Emission Scanning Electron Microscope (FESEM) (Zeiss Merlin-Compact, Oberkochen, Germany) was used to observe the surface morphology of the coated films. Prior to imaging at 3 kV, the samples were mounted with a double-sided adhesive carbon tape on an aluminium stub and vacuum-coated with a thin film of iridium (approximately 5.0 nm) to ensure that the samples were electrically conductive [20].

### 2.8. Functional Groups Profiling: Fourier Transform Infrared-Attenuated Total Reflectance (FTIR-ATR)

The Fourier Transform Infrared-Attenuated Total Reflectance (FTIR-ATR) spectroscopy technique was performed to identify the presence of functional groups in the borosilicate glass and stainless steel substrates before and after the PDMS solution treatment [21]. The FTIR analysis was performed using the FTIR/Fourier Transform Near-Infrared Spectroscopy (FT-NIR) (Perkin Elmer, Beaconsfield, United Kingdom) with a wave-number range of 4000−650 cm^−1^. 

### 2.9. Statistical Analysis

The Design-Expert software version 6.0.10 (Stat Ease, Minnesota, MN, USA) was employed to perform the experimental design and statistical analysis. The collected data were also analysed via the ANOVA and Duncan’s tests using the Statistical Analytical System (SAS®) version 6.12 (SAS Institute Inc., Cary, NC, USA). All experiments were performed in triplicates. The optimum point was validated using the root-mean-squared deviation (RMSD), as shown in Equation (3) [22]:(3)$$\mathrm{RMSD}=\sqrt{\frac{1}{n-1}}\sum_{t=1}{\left(\hat{y}i–yi\right)}^{2}$$

*ŷ_i_* = experimental value;

*y_i_* = expected value;

*n* = number of sample.

## 3. Results and Discussion 

### 3.1. Optimisation of PDMS Treatment on Stainless Steel and Borosilicate Glass 

The observed response variables for the contact angle profiles of the borosilicate glass and stainless steel surface coating treated with PDMS are shown in Table 3. The contact angle data were fitted using linear, quadratic, and cubic models. Based on the statistical results, it is suggested that the modified cubic model with an inverse transformation was the most suitable model for borosilicate glass, while the modified cubic model was the most appropriate for stainless steel substrates. The response surface equation for the contact angle data fitting based on the models is shown in Table 4. According to the variance analysis, both models were significant, where the R^2^ value for both models was higher than 0.75, indicating a good fit. In addition, the lack-of-fit test for both substrates were insignificant, which also demonstrated a good fit between the experimental data and the model. 

Table 5 describes the analysis of the coefficient for each model used to fit the contact angle data for the borosilicate glass and stainless steel substrates. It can be observed from Table 3 that the PDMS concentration significantly affected the contact angle (*p* < 0.05), although the quadratic (x_11_) and cubic (x_111_) effects of the PDMS concentration were negative, which decreased the contact angle. In contrast, the quadratic effect of the PDMS concentration (x_11_) for the stainless steel substrate increased the contact angle, but the interactive effect of both PDMS concentration and treatment time affected negatively the contact angle (x_122_).

Figure 3 shows the three-dimensional imaging of the response surface of the contact angle profiles for the borosilicate glass. Based on the cubic model with an inverse transformation, the contact angle of the borosilicate glass was only affected by the PDMS concentration without any interaction with the treatment time. The highest contact angle was recorded at low PDMS concentrations (1%, *w*/*v*) and decreased further with increasing PDMS concentration. The concentration of PDMS at 1% (*w*/*v*) was sufficient to increase the contact angle surface of the untreated borosilicate glass to become hydrophobic at 90–100° compared to the initial contact angle before the treatment, around 34°. Similar findings were previously reported [23], in which the increase in the silicon concentration through trimethyloxicilicate (TMSS) silicon showed no effect on the contact angle value of the copper substrate. While the surface remained hydrophilic without any TMSS, a small addition of TMSS caused the copper substrate to become superhydrophobic and increased the contact angle. On the other hand, the further addition of TMSS decreases the contact angle value [23]. Likewise, Zhang et al. (2012) [24] applied PDMS to treat the borosilicate crown optical glass surface to make it hydrophobic and increase the contact angle from 23.4° to 75.3°. Through a dehydration process, the PDMS removes the available hydroxyl groups and introduces hydrophobic methyl groups to the coating, thus enhancing the hydrophobicity of the treated glass surface [17,24].

Figure 4 shows the 3D plot of the response surface of the contact angle profiles for the stainless steel substrate. The results showed that both the PDMS concentration and the treatment time did not have a major impact on the surface of the substrate, as shown in Table 5. However, the interactive effect of these two factors, x_1_x_2_^2^, was significant with negative coefficient values (*p* < 0.05). These findings revealed that the increase in PDMS concentration and treatment time did not provide any significant improvement in the contact angle evaluation. In other words, the contact angle of the stainless steel surface showed no substantial increase as the treatment time was prolonged at low PDMS concentrations. In fact, the hydrophilicity of the stainless steel surface was only maintained as the concentration of the PDMS increased during a short treatment time. Hence, the findings suggest that the stainless steel surface’s contact angle value would only increase at low PDMS concentrations and a shorter treatment time.

Additionally, the treated stainless steel surface had a lower degree of increment in contact angle compared to the borosilicate glass. Although the maximum contact angle of both substrates could reach nearly the same at 90–100°, the natural contact angle of the stainless steel surface was higher (75.14°) compared to the borosilicate glass surface (32.51°). The characteristics of the stainless steel hamper the ability of the PDMS solution to raise the contact angle of the stainless steel surface (33% increase) to the same degree as the borosilicate glass surface (200% increase) from its normal contact angle. This was because the formation of the silanol group from the hydrolysis process in the PDMS solution formed an unstable condensation bond with the oxide group of iron or carbon. Moreover, Si-O did not form stable bonds with alkaline metal oxides or carbonates [25]. 

### 3.2. Determination of the Model Validity

The validity of the selected model was evaluated by repeating the tests to acquire the RMSD value, which represents the difference between the predicted and actual optimal points. The repetitive tests were carried out using the optimum parameter values according to the response surface analysis results. The calculated RMSD values using Equation (3) in Table 6 obtained a small RMSD value (1.64) for the glass and (3.37) stainless steel substrates, which validates the selected model.

### 3.3. Surface Morphology of Borosilicate Glass and Stainless Steel Substrates 

Figure 5a,c depicts the surface structure variations of the borosilicate glass before being treated with the PDMS solution and Figure 5b,d after being treated with the PDMS solution as observed under the FESEM at various magnification scales (1000× and 5000× from top to bottom). The observation in Figure 5b reveals that the chemical treatment with the PDMS solution on the borosilicate glass caused a comprehensive morphological change in the substrate surface compared to its natural structure, as shown in Figure 5a. 

The surface of the borosilicate glass was hydrophilic since it was naturally rough with nanostructures. The hydrophilic property is contributed by the highly extensive contact between solids/liquids, which increases with the surface roughness of a material [26]. However, after the glass was treated with the PDMS solution, a layer of hydrophobic material from the PDMS solution covered the glass, which trapped more air on its surface and reduced the solids/liquid contact, resulting in a hydrophobic surface. The previous investigation also used SEM to study the structure of PDMS sheets treated with oxygen plasma and yielded a similar array of nanostructures [26]. 

Aside from the roughness factor, any irregularities, or the presence of holes on the surface should be considered when discussing the surface wetness qualities [27]. Figure 6a,c illustrates the FESEM images of untreated stainless steel surfaces at 1000× and 5000× magnification, while Figure 6b,d shows the FESEM imaging of treated stainless steel surfaces with PDMS solutions. Figure 6a depicts the surface of the natural stainless steel before the treatment, with no PDMS solution applied. The surface morphology of the borosilicate glass treated with the PDMS solution was distinguished by the presence of many grooves and curves, which demonstrated significant surface morphology differences compared to the relatively flattened surface of the untreated borosilicate glass. The previous SEM examination of a zinc-coated steel surface with functional silicon revealed the presence of holes that were etched on the sample surface, allowing the microstructure to trap considerable amounts of air and making the surface superhydrophobic [28]. 

Furthermore, the addition of a thin layer of diluted PDMS solution increased the contact angle value of the stainless steel surface from 75.14° to 104°. The hydrophobic characteristics of the treated surface were based on the Cassie-Baxter model, which states that the low wettability is due to the formation of the rough nano-sized arrangement covered by a hydrophobic substance with air trapped in its topography [29].

### 3.4. Determination of Functional Groups on the Surface of the Borosilicate Glass and Stainless Steel Substrates

The FTIR spectra were examined in the region of 4000–650 cm^−1^ to analyse the changes in the chemical structure of the substrate surface. The FTIR spectra of the untreated borosilicate glass in Figure 7 revealed individual peaks at 767 cm^−1^ and 918 cm^−1^ that correspond to the Si-O-Si and Si-O- groups, respectively [30]. The absence of a peak at 1259 cm^−1^ demonstrated that PDMS was not present on the surface of the untreated borosilicate glass [31]. On the contrary, the obtained spectra on the surface of the borosilicate glass treated with PDMS indicated the presence of additional peaks, notably Si-O-Si (1008 cm^−1^), Si-CH_3_ (1259 cm^−1^), and CH_2_ and CH_3_ (2854 cm^−1^, 2927 cm^−1^, and 2962 cm^−1^) (Ting et al., 2014). 

The considerable drop in the Si-O- concentration peak as well as the increase in Si-O-Si concentration across the surface spectrum of the treated borosilicate glass could be attributed to the changes in the functional group of borosilicate glass surfaces from Si-O- bonds to Si-O bonds [30]. The hydrophobic radicals that caused the surface of the glass substrate to change properties comprising Si-CH_3_, CH_2_, and CH_3_ were formed as the result of the breakdown of the PDMS solution. Consequently, these radicals react with the borosilicate glass to form a hydrophobic coating [2].

The prominent peaks at 1006 cm^−1^ were attributed to the Si-O-Si bonds as a result of the stretching motion of the Si-O-Si bridge. Due to the action of silanol, the peaks were subsequently split into two bands at 1043 cm^−1^ and 1006 cm^−1^, as obtained in the previous work [32]. These two distinct peaks are frequently identified in oligomers, polymers, and PDMS networks [33]. The presence of this peak supports the formation of the Si-O-Si network produced through the addition of the PDMS solution. Based on the FTIR data, it was deduced that the Si-O-Si networks were developed on the surface of the PDMS-treated borosilicate glass, which contributes to the hydrophobic characteristics of the borosilicate glass surface.

As shown in Figure 8, the FTIR-ATR evaluation of the PDMS-treated stainless steel surfaces revealed the spinal structure of the -Si (CH_3_)_2_-O- network, which was indicated in the PDMS spectrum by a band that is typical of PDMS. According to the findings, certain differences in peak heights were visible before and after the PDMS treatment. Several weak peaks were identified at 871 cm^−1^ and 791 cm^−1^ on the PDMS-treated stainless steel surface, which were caused by the Si-C vibration and CH_3_ motion of the SiCH_3_ group, respectively [34]. Similarly, the Si-O-Si band was represented by the peaks at 1095 cm^−1^ and 1017 cm^−1^. Meanwhile, the asymmetrical C-H bending was responsible for the new peak at 1245 cm^−1^, while the apex of the methyl group’s stretching motion was linked with a faint band at 2985 cm^−1^. Following that, a strip at the peak of 3480 cm^−1^ was produced by a hydroxyl group that was not available on the untreated stainless steel surface [35]. All peaks at 791, 1017, 1095, 1245, and 2968 cm^−1^ demonstrated the significant presence of PDMS groups on the surface of the treated substrate [34].

### 3.5. Differences of Contact Angle and Milk Surface Tension

Table 7 shows the results of the full cream milk contact angle readings on the surface of the native and optimally PDMS-treated borosilicate and stainless steel surfaces, as well as surface tension and adhesion work. Due to temperature considerations greatly influencing the surface tension values acquired during the studies, notably for dairy products, the contact angle was measured at 25 ± 1 °C [36].

The contact angle of the full cream milk samples on the glass substrate increased significantly from the native glass to the PDMS-treated glass, as expected, similar to the contact angle observed from the native stainless steel to the PDMS-treated stainless steel. This observed tendency is most likely due to the PDMS-treated substrate having lower surface energy than the native substrate. In general, it is more difficult to wet the surface of low-surface-tension liquids with a low surface energy than solids with a high surface energy (formed by a Van der Wals or hydrogen bond) [37]. As a result, a greater contact angle is generated. 

The surface tension of full cream milk at room temperature was reported at 44 mJ/m^2^ [38]. According to results in Table 7, the surface tension of full cream milk obtained in this study ranged from 48 mJ/m^2^ to 62 mJ/m^2^. Meanwhile, the surface energy of the PDMS-treated borosilicate (25.72 mJ/m^2^) and the PDMS-treated stainless steel (19.13 mJ/m^2^) (Table 7) reveals that all substrates have a surface energy that is lower than the surface tension of full cream milk. As a result, the wettability of the treated substrate is low, resulting in a higher contact angle as compared to the native glass and stainless steel substrates, which have a higher surface energy. 

### 3.6. Differences in Adhesion Work

The strength of the deposition is critical because it determines whether deposits will peel off or slide along the dryer wall once they reach a certain thickness [5]. It also has an impact on the effectiveness of the spray dryer’s cleaning. The adhesion strength of the milk product on the wall of the spray dryer chamber was calculated using Equation (2), which considers the contact angle of the full cream milk sample on the substrate surface as well as the surface tension of the liquid on each substrate. The higher the substrate surface energy and the lower the contact angle of the sample, the greater the force required to separate the liquid from the solid substrate surface [39], whereas previous research on modifying surface energy to reduce the mineral deposition in a milk solution found that deposition on low energy surfaces resulted in a higher adhesion strength [40,41]. According to the results in Table 6, the full cream milk had a higher adhesion work on native glass (81.75 mJ/m^2^), than on the PDMS-treated glass (46 mJ/m^2^). Similarly, when compared to the PDMS-treated stainless steel (41.69 mJ/m^2^), the native substrate had the highest adhesion work (81.75 mJ/m^2^). It will be more difficult to clean the powder adhered to the walls of the drying chamber as the adhesion work increases [1]. This demonstrates that the PDMS treatment on the surface of glass and stainless steel substrates can reduce the adhesion work of milk samples, assisting in the spray dryer cleaning process. 

## 4. Conclusions

This study demonstrated the successful use of a PDMS solution to form a hydrophobic surface on borosilicate glass and stainless steel substrates. The applied surface treatment increased the contact angle between the two surfaces, therefore altering their nature to become more hydrophobic. Based on the RSM analysis, the PDMS concentration had a greater effect compared to the treatment period on the contact angle of the borosilicate glass, while both parameters had a negative interaction effect on the stainless steel substrate. At low PDMS concentrations, both substrates exhibited excellent hydrophobic characteristics, which achieved the highest contact angle increment in the range of 90–100°. In comparison, the PDMS solution was more effective on the borosilicate glass than on the stainless steel surface, as indicated by the percentage increase in the contact angle measurement. The FESEM micrograph in this study also revealed the presence of nanostructures that trap air between the hydrophobic layers, hence boosting the water repellent capacity. Furthermore, the FTIR examination showed the development of new peaks on the substrate surface, notably Si-O-Si, Si-CH_3_, CH_2_, and CH_3_, indicating the presence of interaction between the PDMS solution and the substrate surface. In addition, the surface energy of the substrate surfaces coated with PDMS was notably lower compared to that of the untreated substrates. The whole milk sample, which has a higher surface tension than the treated substrates, exhibited a lower wettability compared to the untreated substrates. As a result, the milk contact angle increased, suggesting that the surface properties shifted from hydrophilic to hydrophobic. As a result, the treated substrate surfaces had reduced adhesion work, making adherent particles easier to clean. Overall, the study presented a simple and economical approach, with no specialised equipment or harsh processing conditions, for an appropriate surface treatment via a spray dryer application to alleviate the product adhesion issues as the transition from hydrophilic to hydrophobic surface qualities was effectively accomplished. 

## Figures and Tables

**Figure 1 molecules-27-03388-f001:**
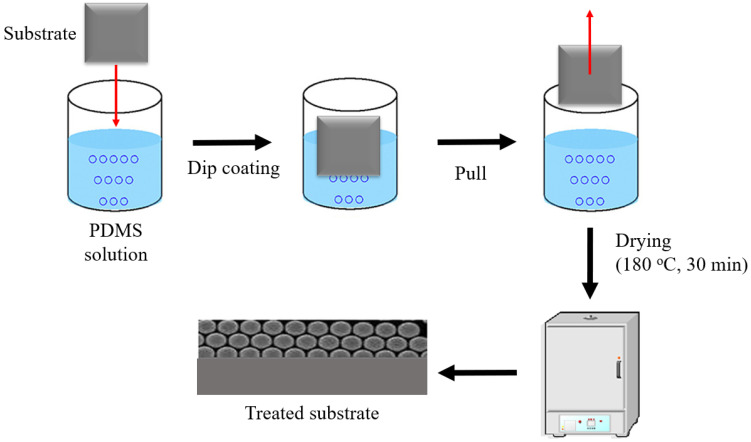
Procedure of the chemical treatment of the substrate surface (borosilicate glass and stainless steel) using the PDMS solution (%, *w*/*v*) via the complete dipping method.

**Figure 2 molecules-27-03388-f002:**
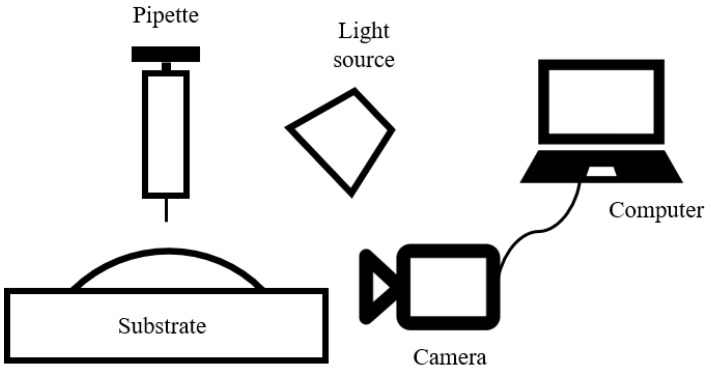
Experimental setup for the static contact angle measurement using the Drop Shape Analyser.

**Figure 3 molecules-27-03388-f003:**
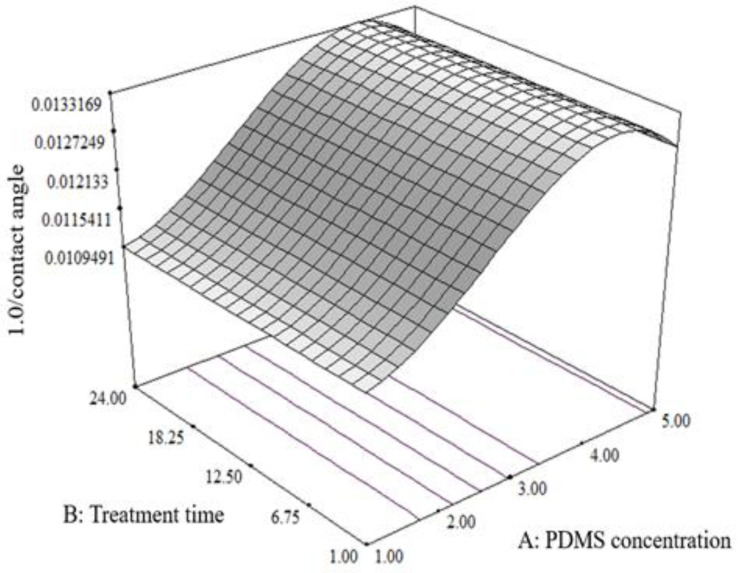
Response surface plot of PDMS concentration versus treatment time on the contact angle value of the borosilicate glass.

**Figure 4 molecules-27-03388-f004:**
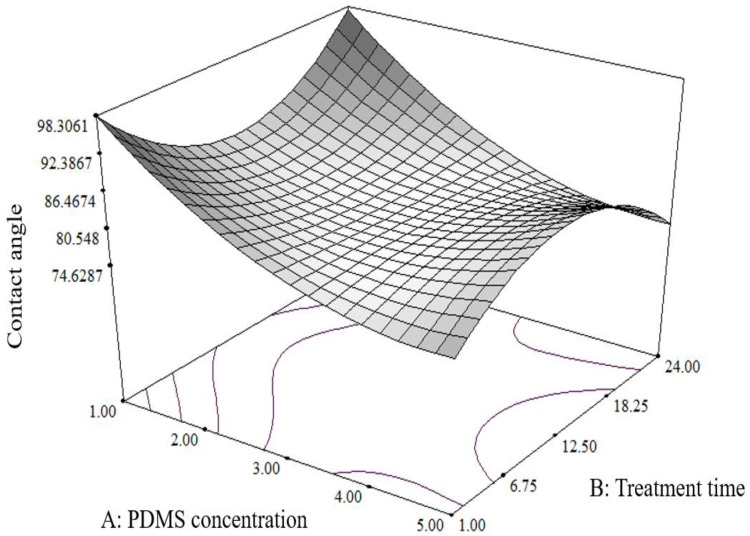
Response surface plot of PDMS concentration versus treatment time on the contact angle value of the stainless steel substrate.

**Figure 5 molecules-27-03388-f005:**
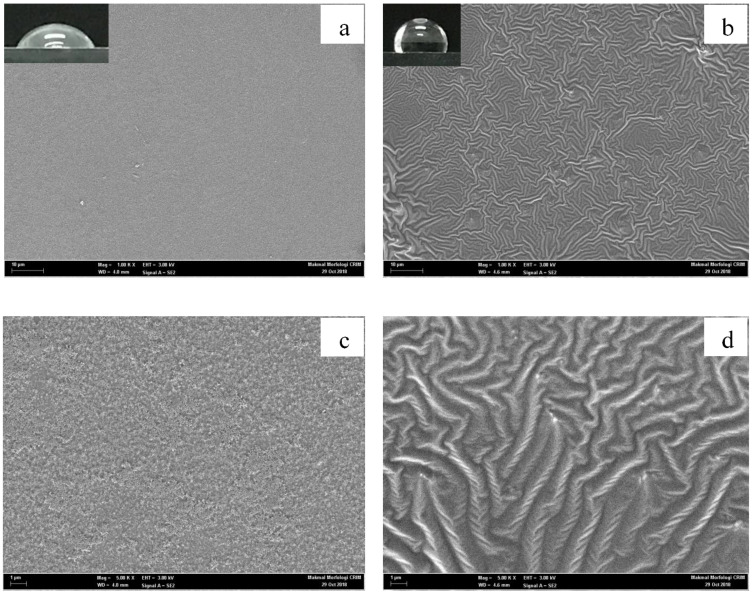
Surface morphology of the borosilicate glass under (**a**,**c**) untreated conditions and (**b**,**d**) PDMS-treated conditions at different magnifications: (**a**,**b**) at 1000× magnification and (**c**,**d**) at 5000× magnification.

**Figure 6 molecules-27-03388-f006:**
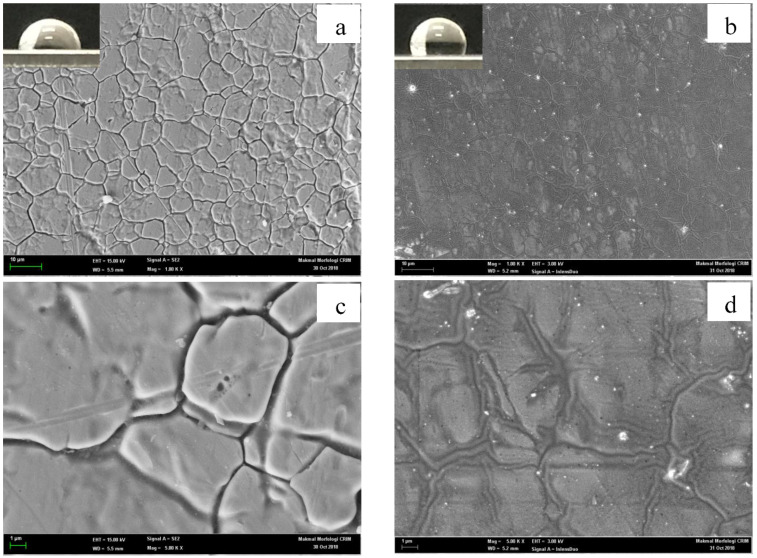
Surface morphology of the stainless steel substrates under (**a**,**c**) untreated conditions and (**b**,**d**) PDMS-treated conditions at different magnifications: (**a**,**b**) at 1000× magnification and (**c**,**d**) at 5000× magnification.

**Figure 7 molecules-27-03388-f007:**
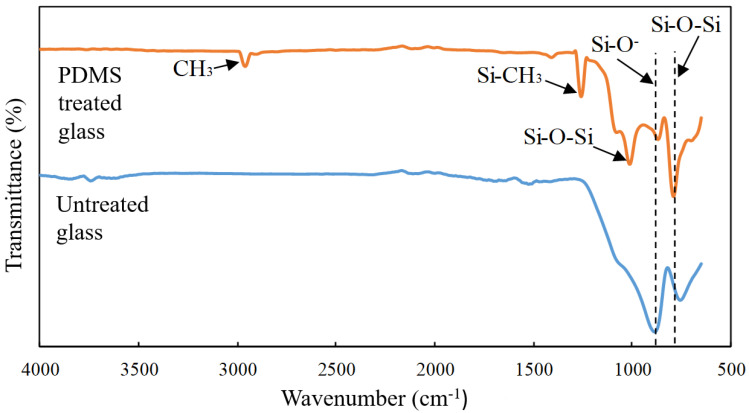
FTIR spectra of the untreated and PDMS-treated borosilicate glass surfaces.

**Figure 8 molecules-27-03388-f008:**
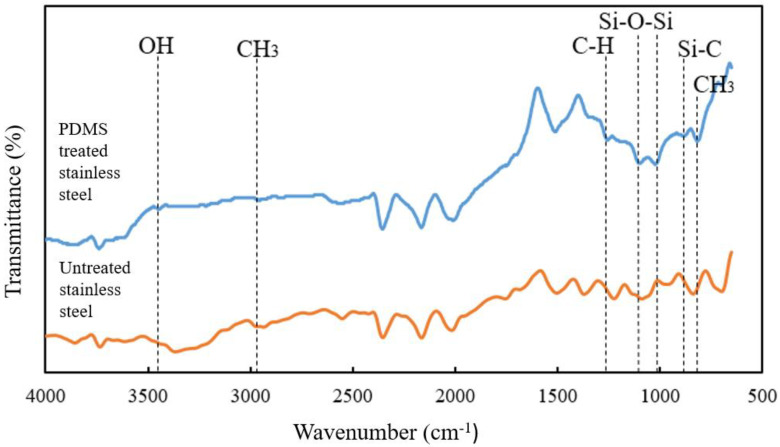
FTIR spectra of the untreated and PDMS-treated stainless steel surfaces.

**Table 1 molecules-27-03388-t001:** Actual and coded (in parentheses) levels of PDMS concentration (X_1_/x_1_) and treatment time (X_2_/x_2_) used for the optimisation of the hydrophobic surface coating process on the borosilicate glass and stainless steel substrates.

Run	X_1_ (x_1_) (%, *w*/*v*)	X_2_ (x_2_) (h)
1	5.00 (1.000)	24.00 (1.000)
2	1.00 (−1.000)	1.00 (−1.000)
3 *	3.00 (0.000)	12.50 (0.000)
4 *	3.00 (0.000)	12.50 (0.000)
5 *	3.00 (0.000)	12.50 (0.000)
6	3.00 (0.000)	−3.76 (−1.414)
7	3.00 (0.000)	28.76 (1.414)
8	0.17 (−1.414)	12.50 (0.000)
9	1.00 (−1.000)	24.00 (1.000)
10	5.00 (1.000)	1.00 (−1.000)
11 *	3.00 (0.000)	12.50 (0.000)
12 *	3.00 (0.000)	12.50 (0.000)
13	5.83 (1.414)	12.50 (0.000)

* Replication of the centre point.

**Table 2 molecules-27-03388-t002:** Composition of the whole milk product.

Composition	Whole Milk (%)
Fat	26
Protein	22.8
Lactose	42.4
Moisture	4

**Table 3 molecules-27-03388-t003:** Actual levels of the independent variables along with the observed values for the response variables, the contact angle of the borosilicate glass and stainless steel treated with PDMS.

Run	X_1_	X_2_	Contact Angle (°) (Y)	
			Borosilicate Glass	Stainless Steel
1	5.00	24.00	78.48	76.72
2	1.00	1.00	90.75	97.82
3 *	3.00	12.50	78.47	77.51
4 *	3.00	12.50	76.74	80.53
5 *	3.00	12.50	80.37	81.23
6	3.00	−3.76	82.5	83.54
7	3.00	28.76	77.02	78.16
8	0.17	12.50	87.27	90.93
9	1.00	24.00	96.69	99.93
10	5.00	1.00	81.14	76.13
11 *	3.00	12.50	77.56	80.93
12 *	3.00	12.50	77.23	78.92
13	5.83	12.50	88.83	91.55

* Replication of the centre point.

**Table 4 molecules-27-03388-t004:** Model equations fitted for contact angle experimental data for PDMS surface coating on borosilicate glass and stainless steel substrates.

Substrate	Response	Model Equation	Model Significance	Lack of Fit	R^2^
Borosilicate glass	Contact angle	Actual equation 1.0/Y = 0.011374 − 1.22587 × 10^−3^X_1_ + 9.25626 × 10^−4^X_1_^2^ − 1.24670 × 10^−4^X_1_^3^ Coded equation 1.0/y = 0.013 + 1.924 × 10^−3^x_1_ − 7.856 × 10^−4^x_1_^2^ − 9.974 × 10^−4^	0.0006 (Significant)	0.0797 (Not significant)	0.8416
Stainless steel	Contact angle	Actual equation Y = 114.65007 − 15.60827X_1_ − 3.42166X_2_ + 1.52731X_1_^2^ + 0.13671X_2_^2^ + 1.06516X_1_X_2_ − 0.043267X_1_X_2_^2^ Coded equation y = 79.82 + 0.22x_1_ − 0.61x_2_ + 6.11x_1_^2^ + 0.91x_2_^2^ − 0.38x_1_x_2_ − 11.44x_1_x_2_^2^	0.0004 (Significant)	0.1216 (Not significant)	0.9644

**Table 5 molecules-27-03388-t005:** Analysis of coefficients for the coded models used to fit the contact angle experimental data for the PDMS surface coating on borosilicate glass and stainless steel substrates.

	Contact Angle		
	Coefficient	F	Prob < F
**BOROSILICATE GLASS**			
*Independent variables*			
PDMS concentration, x_1_	1.924 × 10^−3^	18.10	0.0021
Treatment time, x_2_	-	-	-
*Interactions*			
x_11_	–7.856 × 10^−4^	26.71	0.0006
x_111_	–9.974 × 10^−4^	12.16	0.0069
**STAINLESS STEEL**			
*Independent variables*			
PDMS concentration, x_1_	0.22	0.04	0.8467
Treatment time, x_2_	–0.61	0.64	0.4548
*Interactions*			
x_11_	6.11	55.04	0.0003
x_12_	–0.38	0.12	0.7384
x_22_	0.91	1.23	0.3094
x_122_	–11.44	55.52	0.0003

**Table 6 molecules-27-03388-t006:** Comparison of the expected contact angle readings of the borosilicate glass and stainless steel surfaces with the actual contact angle measured in repeated trials with optimum parameters to determine the model validity.

Substrate	Optimum PDMS Concentration (%, *v*/*v*)	Optimum Treatment Time (h)	Projected Contact Angle Value (°)	Contact Angle Value from the Repeated Experiment (°)			RMSD
				Replication 1	Replication 2	Replication 3	
Borosilicate glass	1	4.92	90.91	88.83	91.83	91.39	1.64
Stainless steel	1	1	98.31	99.82	100.94	101.98	3.37

**Table 7 molecules-27-03388-t007:** The contact angle, surface tension, and adhesion work of whole milk products analysed on the native and PDMS-treated borosilicate glass and stainless steel substrates. Different letters in the same column show significant differences (*p* < 0.05).

Liquid Product	Substrate Surface	Contact Angle (°)	Surface Tension (mJ/m^2^)	Adhesion Work (mJ/m^2^)
Whole milk	Borosilicate glass	68.30 ± 0.38 ^e^	59.68 ± 0.28 ^a^	81.75 ± 0.33 ^a^
	PDMS-borosilate glass	102.81 ± 0.35 ^a^	59.63 ± 0.19 ^a^	46.41 ± 0.21 ^d^
	Stainless steel	82.03 ± 1.30 ^d^	61.90 ± 0.09 ^a^	70.48 ± 1.30 ^b^
	PDMS-stainless steel	103.24 ± 2.50 ^a^	54.11 ± 1.51 ^b^	41.69 ± 1.14 ^ed^

## Data Availability

This study presented an entirely new dataset that is not available anywhere else.

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
