# Peer review of "Response Surface Optimisation of Polydimethylsiloxane (PDMS) on Borosilicate Glass and Stainless Steel (SS316) to Increase Hydrophobicity"

_molecules, 2022, doi:10.3390/molecules27113388_

Round 1

Reviewer 1 Report

This manuscript fabricated Polydimethylsiloxane (PDMS) on borosilicate glass and stainless-steel (SS316) and studied their hydrophobicity, which is meaningfull. In this stuy, PDMS coatings were prepared via the complete dipping method, which is very traditional and lack of novelty. The properties of as prepared PDMS coatings were not outstanding. So, I suggest this manuscript will focus on new preparation mechanism and improving the hydrophobicity and stability of PDMS coatings.

Author Response

Dear reviewers,

Attached herewith are the lists of correction for your kind perusal.

Comments from reviewers and responses to the comments for manuscript, entitled ‘Response Surface Optimization of Polydimethylsiloxane (PDMS) on Borosilicate Glass and Stainless-Steel (SS316) to Increase Hydrophobicity’ are as follows:

Reviewer #1: Reviewer suggest this manuscript will focus on new preparation mechanism and improving the hydrophobicity and stability of PDMS coatings.

Correction and response:

  1. The durability of the coating is deemed to be significant throughout the studies. However, this research aims attention at optimization of coating parameters which give the best possible hydrophobic properties on the borosilicate glass and stainless-steel surface and manifesting the potential concept of having hydrophobic coating on the spray dryer drying chamber.
  2. The result in study proved that the PDMS solution even at low concentration, succeeded to create hydrophobic properties at the optimum time and concentration through optimization process which in accordance with previous research
  3. The author already made some amendment to characterize more on the fundamentals of surface wettability, in term of contact angle, surface energy, surface tension and work of adhesion. Track changes and highlighted changes were carried in the main text as requested.  

Best regards,

Ts. DR. SAIFUL IRWAN ZUBAIRI 

Ph.D (Chemical Engineering) Imperial College London, M.Eng. (Bioprocess), B.Eng. (Hons.)(Chemical-Bioprocess) UTMalaysia | P.Tech. (MBOT), AMIChemE, PMIFT, Grad. (BEM)   

Department of Food Sciences, Faculty of Science & Technology, 

Universiti Kebangsaan Malaysia, 43600 UKM Bangi, Selangor.

+603-8921-5989 | [email protected] | [email protected]

Reviewer 2 Report

Reading through the paper indicates that the paper basically reports a simple method for surface modification based on a common material, PDMS. It is well known that PDMS is hydrophobic and there is no new structures, no physics and no process revealed currently. For cleaning, either superhydrophobic or superhydrophilic surface is used, but not hydrophobic. The authors might want to further explore how to achieve this by structural control. The wettability of surface should be well characterized including the contact angle hysteresis and other (please refer to Bioinspired interfacial materials with enhanced drop mobility: From fundamentals to multifunctional applications). Moreover, the long-term stability of surface property should be demonstrated as well.

Author Response

Dear reviewers,

Attached herewith are the lists of correction for your kind perusal.

Comments from reviewers and responses to the comments for manuscript, entitled ‘Response Surface Optimization of Polydimethylsiloxane (PDMS) on Borosilicate Glass and Stainless-Steel (SS316) to Increase Hydrophobicity’ are as follows:

Reviewer #2: The wettability of surface should be well characterized including the contact angle hysteresis and other (please refer to Bioinspired interfacial materials with enhanced drop mobility: From fundamentals to multifunctional applications). Moreover, the long-term stability of surface property should be demonstrated as well.

Correction and response:

  1. The durability of the coating is deemed to be significant throughout the studies. However, this research aims attention at optimization of coating parameters which give the best possible hydrophobic properties on the borosilicate glass and stainless-steel surface and manifesting the potential concept of having hydrophobic coating on the spray dryer drying chamber.
  2. The result in study proved that the PDMS solution even at low concentration, succeeded to create hydrophobic properties at the optimum time and concentration through optimization process which in accordance with previous research
  3. The author already made some amendment to characterize more on the fundamentals of surface wettability, in term of contact angle, surface energy, surface tension and work of adhesion. Track changes and highlighted changes were carried in the main text as requested.  

Best regards,

Ts. DR. SAIFUL IRWAN ZUBAIRI 

Ph.D (Chemical Engineering) Imperial College London, M.Eng. (Bioprocess), B.Eng. (Hons.)(Chemical-Bioprocess) UTMalaysia | P.Tech. (MBOT), AMIChemE, PMIFT, Grad. (BEM)   

Department of Food Sciences, Faculty of Science & Technology, 

Universiti Kebangsaan Malaysia, 43600 UKM Bangi, Selangor.

+603-8921-5989 | [email protected] | [email protected]

Round 2

Reviewer 2 Report

Now it can be accepted.